# The Impact of Gender and Age Differences and Infectious Disease Symptoms on Psychological Distress in Quarantined Asymptomatic or Mildly Ill COVID-19 Patients in Japan

**DOI:** 10.3390/ijerph19159083

**Published:** 2022-07-26

**Authors:** Keitaro Murayama, Hideharu Tatebayashi, Takako Kawaguchi, Kousuke Fujita, Kenta Sashikata, Tomohiro Nakao

**Affiliations:** 1Department of Neuropsychiatry, Kyushu University Hospital, 3-1-1 Maidashi, Higashi-ku, Fukuoka 8128582, Japan; 2Fukuoka Mental Health Services Center, 3-1-7 Haramachi, Kasuga 8160804, Japan; tatebayashi-h8628@pref.fukuoka.lg.jp; 3Fukuoka City Mental Health Welfare Center, 2-5-1 Maizuru, Chuo-ku, Fukuoka 8100073, Japan; kawaguchi.t04@city.fukuoka.lg.jp; 4Kitakyushu Municipal Mental Health Welfare Center, 1-7-1 Basyaku, Kokurakita-ku, Kitakyushu 8020077, Japan; kousuke_fujita01@city.kitakyushu.lg.jp; 5Graduate School of Human-Environment Studies, Kyushu University, 744 Motooka, Nishi-ku, Fukuoka 8190395, Japan; sashikata.kenta.588@s.kyushu-u.ac.jp; 6Department of Neuropsychiatry, Graduate School of Medical Sciences, Kyushu University, 3-1-1 Maidashi, Higashi-ku, Fukuoka 8128582, Japan; nakao.tomohiro.275@m.kyushu-u.ac.jp

**Keywords:** COVID-19, isolation, psychological distress, K6, mental health, age difference

## Abstract

Quarantine imposed due to COVID-19 infection can exacerbate psychological distress, and it is important for a public mental health agency to identify factors that are predictive of high psychological distress in such situation. The aim of this study was to investigate whether gender, age, and the presence or absence of infectious disease symptoms affected psychological distress among asymptomatic or mildly ill COVID-19 patients who were quarantined. Participants were 436 asymptomatic or mildly symptomatic COVID-19-infected patients who were quarantined in a treatment facility between 1 May 2020 and 30 September 2021. We used Quantification Theory I analysis to investigate the effects of gender, age, and the nature of infectious disease symptoms on psychological distress. The results of the analysis showed that the contribution rate was 0.06. Among gender, age, presence of symptoms, and the nature of symptoms, age had the greatest effect on psychological distress, and being in one’s teens to thirties was considered to exacerbate psychological distress the most. According to the results, the psychological distress of asymptomatic and mildly symptomatic COVID-19 patients isolated was affected by gender, age, and symptomology, especially due to age differences. However, the impact of these items on psychological distress was not considered significant.

## 1. Introduction

A novel coronavirus infection disease (COVID-19) was confirmed in Wuhan, China, on December 2019 and subsequently spread worldwide, with the World Health Organization declaring a pandemic on 11 March 2020. In other countries, urban lockdowns were implemented to prevent infection, and the worsening of mental health due to isolation has been reported, especially among women of child-bearing age [1,2,3,4]. In Japan, lockdowns were not implemented as in other countries, but a “state of emergency” was declared, in which citizens were requested to refrain from an array of daily activities. For example, the government urged people to refrain from using public transportation, and schools switched from face-to-face to online classes. After implementing these measures to prevent the spread of infection, an increase in the risk of suicidal behavior among the general population, especially among women and care-givers, was reported in Japan [5,6,7], and an increase in the total number of suicides [8], especially among women [9,10], was also observed.

While the Japanese government is urging the general population to refrain from such lifestyle activities, in accordance with Japanese law (Infectious Disease Control Law), asymptomatic pathogen carriers and mild COVID-19 patients (those who do not have pneumonia and do not fall into the following four categories: elderly, those with underlying diseases, immunosuppressed, and pregnant) will, in principle, be quarantined in facilities prepared by local governments (prefectures, cities with public health facilities, and special wards) beginning in April 2020. Those admitted to the facilities were isolated in a private room in an accommodation facility and prohibited from direct contact with others until they fulfilled the criteria for discharge (Appendix A). Although this isolation of infected individuals is thought to prevent the spread of infection, effects on the mental health of the individuals involved, such as depression and anxiety, have been reported [11]. This meta-analysis reported that, compared to controls, infected individuals who were isolated were more likely to have depression (odds ratio 2.795; 95% confidence interval 1.467–5.324), anxiety disorders (odds ratio 2.0; 95% confidence interval 0.883–4.527), and stress-related disorders (odds ratio 2.742; 95% confidence interval 1.496–5.027).

In Japan, Mental Health Welfare Centers are public institutions responsible for mental health. These centers are located in each of Japan’s 47 prefectures and 22 designated cities, and they provide key mental health welfare activities in their respective regions by disseminating knowledge, conducting surveys and research, and providing consultation and guidance concerning mental health and the welfare of those with mental disabilities. The staff comprises psychiatrists, nurses, psychologists, and mental health workers. However, because it is not a medical institution, it does not provide drug therapy or psychotherapy, and those who need treatment are referred to a psychiatric institution. Fukuoka Prefecture, with a population of about 5 million, is one of the largest metropolitan areas in Japan after Tokyo, Osaka, and Aichi Prefecture, and three mental health welfare centers have been established here. These mental health welfare centers provide on-site mental health counseling to COVID-19-infected residents of Fukuoka Prefecture who have been quarantined in facilities.

The purpose of this study was to investigate the attributes that influenced psychological distress in COVID-19 patients (hereafter referred to as infected patients) who were isolated in residential treatment facilities and consulted at a mental health welfare center. Because previous studies have reported higher anxiety in women than in men among patients with COVID-19 who were admitted to a medical facility for treatment [12,13,14,15,16], we hypothesized that being female would have an exacerbating effect on psychological distress. Furthermore, given that younger age groups were more anxious under urban lockdowns in overseas reports [1,2,17,18], we postulated a second hypothesis: that being younger than 40 years old would have had an exacerbating effect on psychological distress. The third hypothesis was that the presence of physical symptoms, such as a headache, would have an exacerbating effect on psychological distress since the previous studies showed that the patients with COVID-19 symptoms including headaches or dizziness were more likely to have psychological symptoms [19] and because the psychological distress in patients with COVID-19 correlated positively with a peripheral inflammatory indicator [20].

## 2. Methods

### 2.1. Participants

The participates in this study were the patients who were admitted to residential treatment facilities in Fukuoka Prefecture from 1 May 2020 to 30 September 2021. The inclusion criterion was the patients who requested and received consultations conducted by the three mental health welfare centers in Fukuoka Prefecture. We excluded the patients with missing data for any of the survey items listed Section 2.4.

### 2.2. Survey Methodology

The consultation was conducted by the staff of the mental health welfare center using the telephone in the residential treatment facility in which the patient was admitted. The patient who received consultation was recorded on the individual form which included the survey items by the staff of the mental health welfare center.

The authors (H.T., T.K., and K.F.) retrospectively examined the survey items in the individual form for this study from 1 February 2022 to 21 February 2022.

### 2.3. Measuring Psychological Distress

The Kessler Screening Scale (K6) was used to measure psychological distress [21]. This scale consists of six self-administered questions (“Do you feel irritable?”, “Do you feel hopeless?”, “Do you feel fidgety and restless?”, “Do you feel depressed and uncomfortable about everything that happens?”, “Do you feel like everything you do is a struggle?”, and “Do you feel that you are worthless?”). Each item is rated on a scale of 0 to 4, with the total score ranging from 0 to 24; the higher the score, the greater the psychological distress. It is an excellent screening tool for mood and anxiety disorders [22,23], and the validity of the Japanese version has been confirmed [24].

### 2.4. Survey Items

K6 score, gender, age, and symptoms of infection (“fever”, “headache”, “malaise”, “symptoms of upper respiratory inflammation”, “abnormal sense of taste”, and “abnormal sense of smell” as major symptoms; up to three symptoms per participant) were selected. We defined symptoms of upper respiratory inflammation as nasal obstruction, nasal discharge, cough, sore throat, and phlegm. If no symptoms due to infection were observed, the participant was defined “asymptomatic”.

### 2.5. Statistical Analysis

The test was conducted using IBM SPSS Statistics version 28.0.1.0 (IBM, Armonk, NY, USA). A gender difference in mean K6 score was tested using an independent *t*-test. We conducted the one-way analysis of variance (ANOVA) to investigate the difference between the mean K6 score among the age groups. If there was a significance difference of mean K6 scores among the age groups, a post-hoc test using the Shaffer test was conducted.

Quantification I analysis uses categorical data (qualitative data) as explanatory variables to explain or predict the values of quantitatively measured objective variables. The aim of the current study was to investigate how much each of the categorical data, such as gender, age, and presence of each infection symptom, affected K6, which was the quantitative variable. We therefore used this statistical method. In this study, the explanatory variables were attribute categories such as gender (male and female), age (teens, 20s, 30s, 40s, 50s, 60s, 70s, and above), nature of infection symptoms (asymptomatic, fever, headache, fatigue, symptoms of upper respiratory inflammation, abnormal sense of taste, and abnormal sense of smell), and the K6 score as the objective variable.

### 2.6. Ethical Consideration

This study was approved by the Ethical Review Committee for Observational Research of the Kyushu University Medical School District Office (Permit No. 21118-00). We announced this study on the website of the mental health and welfare centers. We asked those who did not wish to participate in the study to contact us. The personal information used in the study was anonymized and kept at Kyushu University and each mental health and welfare center.

## 3. Results

### 3.1. Participant Background

The number of the patients admitted to residential treatment facilities during the period of the study was 26,071. A total of 518 participants, which is 1.99% of the patients who were admitted to residential treatment facilities, were received through consultations; after excluding 81 participants with missing K6 data and one with missing age data, 436 participants were included in this study.

The backgrounds of the participants are shown in Table 1. The gender breakdown was 184 men and 252 women. By age group, 13 (2.98%) were in their teens, 64 (14.96%) were in their 20s, 80 (18.34%) were in their 30s, 108 (24.77%) were in their 40s, 97 (22.24%) were in their 50s, 45 (10.32%) were in their 60s, and 29 (6.65%) were in their 70s and above. The breakdown of infection symptoms was: asymptomatic in 121 (27.75%), fever in 119 (27.29%), headache in 45 (10.32%), malaise in 59 (13.53%), symptoms of upper respiratory inflammation in 158 (36.23%), abnormal sense of taste in 47 (10.78%), and abnormal sense of smell in 41 (9.40%).

In the mean K6 score, there was no significant difference between men and women (*t*(434) = 0.066, *p* = 0.948). The one-way ANOVA detected significant differences between ages (*F*(6.429) = 3.79, *p* = 0.001). The mean K6 scores of patients in their 20s and the 30s were significantly higher than the mean K6 score of patients in their 70s and above (*p* = 0.001). The mean K6 score of the patients in their 40s was also higher than those in the 70s and above (*p* = 0.025).

### 3.2. Analysis by Quantification I


The results from the Quantification I analysis are shown in Table 2. The category quantity is an indication of how much each category affects the objective variable. In this study, a negative category quantity is predicted to decrease the K6 score. Range is the value obtained by subtracting the minimum value from the maximum value of the category quantity. The higher the value of the range, the greater the difference within the attribute category.

In terms of gender, the category quantity for “female” was −0.135, and that for “male” was 0, suggesting that being female had an effect on decreasing the K6 score. The category quantities for each age group were 1.625 for “teens”, 1.183 for “20s”, 1.130 for “30s”, 0 for “40s”, −0.177 for “50s”, −1.057 for “60s”, and −3.444 for “70s and above”. The results suggest that being in one’s teens to 30s had an increasing effect on the K6 score, while being in one’s 50s or older had a decreasing effect on the K6 score. The category quantities in the attribute category of “Description of infection symptoms” were 0.270 for asymptomatic, 0.329 for fever, 1.368 for headache, −0.692 for malaise, 0.425 for symptoms of upper respiratory inflammation, 0.006 for taste abnormality, and −0.199 for olfactory abnormality.

The maximum range in the attribute category was 5.069 for “age group”, 2.060 for “Symptoms of infection”, and 0.315 for gender.

The multiple correlation coefficient, contribution ratio, and residual standard deviation were 0.245, 0.06, and 5.251, respectively.

## 4. Discussion

The current study investigated the affected psychological distress as measured by the K6 in COVID-19 patients who were isolated in residential treatment facilities. The results showed that women or patients in the 50s and above age group had lower K6 scores. In the “nature of infection symptoms”, the K6 score was higher for participants with “headache”, “fever”, and “symptoms of upper respiratory inflammation” than for those who were “asymptomatic”.

The present study revealed that “being a woman” was not an attribute category that exacerbated psychological distress. However, psychological distress in daily life is higher for women compared to men [25,26,27,28,29], and social factors such as biological [30], psychological, and social roles along with societal gender-based expectations have been cited as causes [31]. In addition, mental disorders such as depression, anxiety disorders, and posttraumatic stress disorder are reported to be more prevalent in women [32,33,34,35]. In the general population during the COVID-19 pandemic, women were more affected by COVID-19-related fear and anxiety than men [36,37], and had more suicidal ideation [37,38] and proclivity to suicide [39]. In addition, the women with COVID-19 were more prone to anxiety and depression than men [13,14,16]. Although these previous studies support the preliminary hypothesis of this study, the results of the present study are contrary to the hypothesis. A study that investigated the gender difference in COVID-19 attitude and behavior revealed that women were more likely to agree with restraining public policy measures and comply with them [40]. It, therefore, is possible that women suffered less psychological distress than men since women might be more accepting of being isolated by Japanese government’s policy than men. Additionally, there is a study which investigated hospitalized COVID-19 patients that suggested that male gender was associated with anxiety [41]. In Italy, men had an increased perception of loneliness compared to women during the COVID-19 pandemic lockdown compared to before it [42]. Moreover, there are some studies which showed that rates of suicidal ideation in men were higher than in women [43,44,45]. These findings might support the results of this study.

In the present study, it was found that, in terms of age generation, those in their teens to 30s had higher K6 scores than those in their 40s and above. This result is consistent with the findings that psychological distress due to the spread of COVID-19 infection is stronger in young adults [17,46,47] and that young adults experience more depression and anxiety under urban lockdown to prevent infection [48,49]. Psychological resilience is defined as the process of successfully adapting to difficult situations and negative stressors through psychological, emotional, and behavioral flexibility and adaptation to environmental and internal demands (American Psychological Association, https://www.apa.org/topics/resilience, accessed on 5 May 2022). Psychological resilience is generally reported to be lower in young adults than in older adults [50]; Gooding et al. reported that older adults have higher levels of emotional regulation and problem solving. In other words, a low tolerance for uncertain situations among young adults [17] and low psychological resilience may have contributed to the higher psychological distress among young adults.

Among the categories affiliated with “symptomatology”, the results of this study supported our hypothesis that those with “fever”, “headache”, and “upper respiratory symptoms” had higher K6 scores than those who were “asymptomatic”. However, the lowering effect of “fatigue”, “taste abnormality”, and “olfactory abnormality” on K6 scores compared with “asymptomatic” participants was contrary to our prior hypothesis. A study of 109 individuals with suspected COVID-19 infection in Taiwan [51] reported that those with psychological distress were significantly more likely to have cough and general malaise when compared to those without psychological distress. However, in this study, only four of the participants ultimately had COVID-19, and only one of the 109 participants had abnormal taste or sense of smell. In other words, the participants in the present study differed from those of previous studies in that they were not in a situation of known COVID-19 infection and a lower percentage of them had taste or olfactory abnormalities. Therefore, further investigation is needed to determine how the presence or absence and nature of symptoms affect the psychological distress of infected individuals.

The results revealed that the range was 5.069 in the attribute category of “age”, which revealed that “age” has the largest effect on the K6 scores among the attribute categories investigated in this study. However, the results of the present analysis showed a contribution rate of 0.06, indicating that it is difficult to predict K6 scores based on “gender”, “age group”, “presence of symptoms”, and “nature of symptoms. The results indicated that other factors such as low income, unemployment, and chronic disorders [4] might have affected psychological distress.

This study has some limitations. The first, as mentioned above, is the limited number of survey items. Items such as history of physical illnesses such as respiratory disease, diabetes, and hypertension, a history of psychiatric illness such as depression and anxiety disorders, family status, income status, whether they were students or employed, how much they communicated with others on the Internet, and social media usage may have affected the K6 score. The contribution rate of the research items investigated in this study was low, at 0.06. Second, K6 was the only psychological assessment scale used in this study, and the Generalized Anxiety Disorder Scale (GAD-7) and the Patient Health Questionnaire (PHQ-9) could be added to the K6 to provide a more detailed understanding of the participants’ psychological states, such as anxiety and depression. Third, the participants were not all those who were isolated in the treatment facilities. The participants of this study were those who requested consultation conducted by the Mental Health Welfare Center, and it is thought that there was a bias toward those with high K6 scores. Therefore, it is possible that the infected persons who were not the participants of this study had lower K6 scores than the participants of this study, and that the results of this study would have been different if those persons were included in the study.

## 5. Conclusions

This study found that the intensity of psychological distress among quarantined individuals infected with COVID-19 can be predicted to be higher among males, among those aged in their teens to thirties, and among those with fever, headache, and symptoms of upper respiratory inflammation than in asymptomatic people. We should pay attention to these items when we conduct consultations with COVID-19 patients who are isolated in a facility.

## Figures and Tables

**Table 1 ijerph-19-09083-t001:** Demographic characteristics of the study sample (*n* = 436).

	Number of People (%)	Mean K6 Score (S. D.)	Statistics
Total	436	11.52 (5.32)	
Men	184 (42.2)	11.54 (5.51)	*t*(434) = 0.066, *p* = 0.948
Women	252 (57.8)	11.50 (5.18)	
Age group			
Teens (10–19)	13 (2.98)	12.92 (7.90)	*F*(6, 429) = 3.79, *p* = 0.001, 70s and above <20s, 30s, *p* = 0.001, 70s and above <40s, *p* = 0.025
20s	64 (14.67)	12.75 (5.00)	
30s	80 (18.34)	12.53 (4.97)	
40s	108 (24.77)	11.50 (5.15)	
50s	97 (22.24)	11.29 (5.44)	
60s	45 (10.32)	10.36 (4.73)	
70s and above	29 (6.65)	8.03 (4.55)	
Symptoms of infection			
Asymptomatic	121 (27.75)		
Fever	119 (27.29)		
Headache	45 (10.32)		
Malaise	59 (13.53)		
Symptoms of upper respiratory inflammation	158 (36.23)		
Abnormal sense of taste	47 (10.78)		
Abnormal sense of smell	41 (9.40)		

**Table 2 ijerph-19-09083-t002:** Quantification I based classification of K6 scores (*n* = 436).

	People	Category Quantity	Range
Gender			0.315
Men	184	0	
Women	252	−0.315	
Age group			5.069
Teens (10–19)	13	1.625	
20s	64	1.183	
30s	80	1.130	
40s	108	0	
50s	97	−0.177	
60s	45	−1.057	
70s and above	29	−3.444	
Symptoms of infection			2.060
Asymptomatic	121	0.270	
Fever	119	0.329	
Headache	45	1.368	
Malaise	59	−0.692	
Symptoms of upper respireory inflammation	158	0.425	
Abnormal sense of tasete	47	0.006	
Abnormal sense of smell	41	−0.199	
Multiple correlation coefficient		0.245

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
