# Peer review of "The Impact of Gender and Age Differences and Infectious Disease Symptoms on Psychological Distress in Quarantined Asymptomatic or Mildly Ill COVID-19 Patients in Japan"

_ijerph, 2022, doi:10.3390/ijerph19159083_

Round 1
Reviewer 1 Report
Dear authors, thank you for allowing me to review this manuscript (IJERPH-1805843)
Below, I provide some comments to help improve the reporting of this research. I hope it helps.
Abstract:
- I would not start this section speaking about depression and/or anxiety since these are major mood disorders, and the authors are talking about psychological distress that does not necessarily means or end up as a depressive or anxiety disorder.
- I think it would be nice to emphasise further this research's needs and implications in the abstract.
Introduction:
- Some wording needs revision (p.e. deterioration of mental health, intense psychotherapy, etc.)
- Information on the K6 (page 2, lines 70-79) would be better placed in the methods section. Now it is placed in the middle of the introduction and two paragraphs, interfering with the logical course of the narrative.
- Information about ethics (page 3, lines 97-98) would be better placed in the article's methods section.
Methods:
- It would be nice to specify potential participants (first section on the methods) from the recruitment centre and % of people declining to participate in the survey (response rate). It is not clear if this information is provided in the results (lines 124-125)
- The individual forms made in the above consultations were retrospectively examined by who?
- Survey items are insufficiently described. For example, the survey as supplementary material could be provided, or a complete description of variables assessed in this study is needed.
- Statistical analyses are insufficiently described (tests employed, level of significance, post-hoc corrections, if any, etc.)
The rationale for using Quantification I and not other statistical tests with quantitative variables (age, number of symptoms, etc.) should be provided.
Results:
- Mean age and proper descriptives could be provided. Also, spelling and typos could be improved when reporting results and in Tables.
Discussion:
- It is not entirely true that this is the first study to investigate the influence of these factors on psychological distress in asymptomatic and mildly ill infected patients who were quarantined since some research has reported mental health effects of quarantine and isolation in different populations (The Lancet has a recent and fascinating study on this topic). An update of references must be provided, and perhaps the stress should be placed on the specific characteristics of the sample. To do so, a more in-depth explanation of the sample should be provided in the results section.
- First lines in the Discussion (from the beginning to 180) are mainly a repetition of the results section. However, I believe a more insightful discussion is needed to avoid duplication and use recent to compare and discuss the research findings.
- I do not understand why authors define and refer to psychological resilience in the discussion since it is not measured or assessed in the research design. It could imply that preventive intervention measures should be designed to address psychological distress in similar situations. However, now it is disconnected from the general line of the Discussion they present. This part should be revised and better explained.
- I see no implications or potential implications based on this research and its findings mentioned in the Discussion or the Conclusions sections.
References:
- References are pretty updated, but some recent research on the psychological effects of quarantine and isolation on different populations and quarantine individuals in different facilities should be added in the Introduction and discussion sections. Perhaps this research was carried out and the manuscript prepared in a time before the boom of publications on this topic, but now, a more extensive corpus of research on these issues exists and must be referred to.
Reviewer 2 Report
That social isolation due to Covid-19 has caused mental illness in many people is a serious, international issue. It is new and requires research if victims are to be helped and further victims prevented. This is a well-structured, methodologically sound exploratory study of the problem in Japan. Some of the results are unexpected, in particular that women are less vulnerable than men. The researchers also discover significant age differences and also that certain covid symptoms are more closely associated with mental problems than others. They make interesting suggestions about the influences upon these differences, but they are clear that this is an experimental study in a previously untried field and it has limitations, which they clearly explain. The value of the article is that it opens up an important new field of research, raising significant issues which will hopefully be followed up more fully in future research.
The main question addressed by the research was to ask what was the nature of the impact of isolation due to Covid-19 on mental illness.
The research was 'new' and important given international concern about this issue and in order to develop ways to help victims and prevent future problems.
I point out that the research reveals important and unexpected age and gender differences in the impact of isolation.
If I had improvements to the methodology to suggest I would have suggested them. I point out that the authors are clear that their work has limitations since it is an experimental study in a new field. They are clear and convincing about these limitations.
The conclusions are consistent with the evidence and arguments and address the main questions posed.
The references are appropriate.
The tables and figures seem good to me.
Reviewer 3 Report
Dear Author:
This study aimed to investigate the attributes that influenced psychological distress measured by K6 in COVID-19 patients, who were isolated in residential treatment facilities and consulted at a mental health welfare center. This study is interesting, appropriate for the current time, and suitable for the theme of this journal. It may be a stimulating paper for the readers of this journal, who are particularly interested in learning about the psychological distress in COVID-19 patients. However, certain points should be modified. Therefore, please make the necessary revisions by referring to the following.
In the lines 62-69 and 80-84, since the explanations are about the mental health and welfare centers nationwide and in Fukuoka prefecture, respectively, it would either be good to continue these as two paragraphs or can be combined into one paragraph.
In the lines 78-79, the author presents a K6 cutoff point; however, its relation to the present study remains unclear. It seems that the K6 cutoff value has not been set in this study.
In the lines 88-96, the author presents three hypotheses; however, only one document (reference no. 13) in support of the first hypothesis was cited (too few). Provision of more relevant literature is requested.
In the lines 94-96, the author presents a third hypothesis without citation of supporting literature. Some relevant literature should be cited.
In the lines 97-98, a statement about the ethical consideration is provided; however, it should be mentioned elsewhere other than the end of the introduction section.
In this study, description of obtaining the consent of participants or protecting personal information is absent; whereas it should be described.
Regarding the results of K6 scores, although the mean score and standard deviation are described in Table 1, they do not match the item name “Number of people (%)”; therefore, a presentation should be devised. For the results of K6 scores, not only the mean and standard deviation but also other more detailed data should be provided.
In the lines 164-165, a case of a “female has an effect of reducing psychological distress, which is contrary to our hypothesis” is described. Although the results are contrary to several previous results, some supportive results, even though few, should have been provided. Search and citation of past literature is advised, that states that males scored higher than females when the K6 results were based on gender.
In the lines 195-198, the author describes possible reasons for the results of this study based on gender differences. The author presents “receive more social support through social networking services than men” and “received more social support via the Internet” as reasons. This reason is incompatible with many other reports that stated that women’s stress and suicide have increased because of the COVID-19 pandemic; therefore, further consideration is required. Considering the author’s reason is correct, regardless of this study, women’s stress and suicide did not increase in the COVID-19 pandemic as compared to men. This is because women were able to deal with it utilizing SNS and the Internet, even if they are forced to refrain from doing so or were restricted during the COVID-19 pandemic. Therefore, this view contradicts the first hypothesis of this study. Regarding this point, the author should reconsider.
Additionally, more literature should be cited which states that COVID-19 pandemic has increased women’s stress and suicide.
In the lines 226-242, the author describes three limitations of this study. Regarding the second limitation, the meaning of the description “The second is that the psychological evaluation scale had a significant effect on K6” is unclear. Appropriate corrections are required. Moreover, the word "second" is repeated (The second is… on K6. Second, …); therefore, please correct them as required.
Regarding the conclusion, it should always be derived from the results and considerations up to the previous stage. In the lines 248-250, the parts of the results and discussions from which the description (Investigations into the factors … should include social factors.) was derived, remains unclear. Please make appropriate corrections to clarify the relevance of the conclusion to the previous descriptions.
Round 2
Reviewer 1 Report
Dear authors, thanks for considering all the comments and taking the time to improve your research. Most issues raised have been resolved. I still feel that the methods section must be improved by providing a clear structure, organization and more details about the procedure (now it is under survey methods? Just two sentences) I consider a more traditional structure of methods would help readers to find relevant information and concerning procedure, the how, when, who, length of the assessments, for instance, could be provided.
Before final publications, a general English check must be performed too.
Good luck with your research. Sincerely,
Reviewer 3 Report
Dear author:
I have confirmed that you have carefully examined the matters I pointed out in my previous comments and made the necessary corrections to improve your manuscript. I find the manuscript suitable for publication. Thank you for responding appropriately to my requests. This paper studied the attributes that influenced psychological distress in COVID-19 patients, who were isolated in residential treatment facilities and consulted at a mental health welfare center. This study is timely and interesting and is in line with the theme of this journal and can be said to be a stimulating paper for the readers of this journal. This study has several useful contributions for future research.
